# Augmented Mixup Procedure for Privacy-Preserving Collaborative Training

## Abstract

Mixup, introduced by Zhang et al., is a regularization technique for training neural networks that generates convex combinations of input samples and their corresponding labels. Motivated by this approach, Huang et al. proposed InstaHide, an image encryption method designed to preserve the discriminative properties of data while protecting original information during collaborative training across multiple parties. However, recent studies by Carlini et al., Luo et al., and Chen et al. have demonstrated that attacks exploiting the linear system generated by the mixup procedure can compromise the security guarantees of InstaHide. To address this vulnerability, we propose a modified mixing procedure that introduces perturbations into samples before forming convex combinations, making the associated linear inverse problem ill-conditioned for adversaries. We present a theoretical worst-case security analysis and empirically evaluate the performance of our method in mitigating such attacks. Our results indicate that robust attack mitigation can be achieved by increasing the perturbation level, without causing a significant reduction in classification accuracy. Furthermore, we compare the performance of our approach with that of InstaHide on standard benchmark datasets, including MNIST, CIFAR-10, CIFAR-100, and Tiny-ImageNet.

## 1 Introduction

Data mixing was initially introduced as a dataset augmentation technique, generating new samples by computing weighted averages of subsets from the original dataset Zhang et al. (2017). Originally designed as a regularization method for training neural networks, this approach has also been adapted for privacy-preserving protocols, as the mixing process obscures the original data during model training Liu et al. (2019); Fu et al. (2019).

Although the mixup strategy appears to preserve privacy without significantly degrading model performance, directly applying the method proposed by Zhang et al. (2017) can introduce vulnerabilities that allow attackers to recover private data under certain conditions Huang et al. (2020):

1. **Mixup samples from a private dataset only:** If mixup samples are generated exclusively from a private dataset, an attacker can identify which samples share a common private component by analyzing the expected value of the dot product between mixup samples. Once a set of related mixup samples is identified, the common private sample can be reconstructed by averaging these samples.

2. **Mixup samples from both private and public datasets:** When mixup samples are generated using both private and public datasets, repetitions of private samples in the mixup process can be avoided by leveraging the public dataset. However, since the public dataset is accessible, an attacker can perform a similar statistical analysis to identify which public samples were used in the mixup. Once the public components are determined, the private sample can be trivially reconstructed.

Building on their security analysis, Huang et al. (2020) proposed a mixup-based algorithm called InstaHide. The key innovation of their method is the application of a sign-flipping mask to images generated by computing a weighted sum of both public and private samples. The authors analyzed the security of InstaHide and formally proved that its security depends on the computational hardness of the subset-sum problem. However, the assumptions underlying their security model do not

accurately reflect the properties of real-world data. In particular, Huang et al. (2020) assumed that each sample consists of an arbitrary sequence of values. For example, in the context of images, this assumption implies that pixels are independently and randomly distributed, which does not hold in practice. Consequently, although the security proof is mathematically valid, it does not offer practical security guarantees. This limitation was highlighted by Carlini et al. (2021a), who developed efficient attacks on samples generated by the InstaHide algorithm, enabling near-complete recovery of the original data.

## 2 CONTRIBUTIONS

The primary contribution of this paper, presented in Section 3, is a singularized mixup algorithm that resolves the core weakness exploited by major attacks on InstaHide: the repeated reuse of the same private sample across mixtures. Our method mixes only two private images at a time and injects structured noise into the non-target component, thereby eliminating the persistent signal required for reconstruction attacks.

In Section 4, we develop a theoretical security analysis of the proposed mechanism. We establish lower bounds on the achievable reconstruction error under an adversary with full knowledge of the mixing weights, and we provide a principled way to choose the noise norm so that the SNR associated with separating each encoded sample into signal and interference components remains below a prescribed security parameter $\tau$.

Section 5 presents an empirical study of both security and accuracy. We evaluate linear and nonlinear attackers in a conservative threat model and show that, above a modest noise threshold, neither can recover the underlying private images. At the same time, even when using a noise level far stricter than what the theoretical bounds require, our method maintains accuracy comparable to InstaHide's strongest $k = 4$ configuration on MNIST, CIFAR-10, CIFAR-100, and Tiny-ImageNet. Overall, our results demonstrate that the proposed mixup procedure provides strong protection against inversion while preserving high downstream utility.

## 3 SINGULARIZED MIXUP

In this section, we introduce our algorithm, which is based on the singularization framework. This approach enhances security by ensuring that each execution is unique, thereby making it broadly applicable to a variety of systems Gaber et al. (2023). Singularization has previously been used to strengthen encryption algorithms without modifying their underlying structure Macario-Rat & Plesa (2024), which motivates our adoption of this framework in the design of our mixup algorithm.

We begin with a brief overview of InstaHide, followed by a detailed description of our proposed algorithm.

### 3.1 INSTAHIDE ALGORITHM

Consider a private dataset $(x_i, y_i)_{i=1}^n$ consisting of $n$ samples, where $x_i \in \mathbb{R}^d$ denotes the input example and $y_i \in \mathbb{R}^c$ is the corresponding one-hot encoded label.

The fundamental idea behind Mixup, as introduced by Zhang et al. (2017), is to replace each data point with a convex combination of the current sample and $k - 1$ other samples selected uniformly at random from the dataset. Specifically, each new data point is generated by taking a weighted average of $k$ instances and their associated labels:

$$\tilde{x}_i \leftarrow w_{i1}x_i + \sum_{j=2}^{k-1} w_{ij}x_{\pi_i(j)} \tag{1}$$

$$\tilde{y}_i \leftarrow w_{i1}y_i + \sum_{j=2}^{k-1} w_{ij}y_{\pi_i(j)} \tag{2}$$

where $\{(\tilde{x}_i, \tilde{y}_i)\}_{i=1}^n$ represents the encoded dataset and $\pi_i$ is a random permutation over $\{1, 2, \ldots, n\}$.

The InstaHide approach builds upon Mixup but introduces two key modifications:

1. **Public images**: InstaHide augments the private dataset with samples from public datasets, expanding the pool of mixing samples to $(x_i, y_i)_{i=1}^n \cup (x_i, y_i)_{i=n+1}^{n+m}$, where $m$ denotes the size of the public dataset.

2. **Sign mask**: The sign of each pixel in a mixup image is randomly flipped using a random sign mask $\sigma_i \sim \Lambda_\pm^d$.

As a result, equations (1) and (2) are modified as follows:

$$\tilde{x}_i \leftarrow \sigma_i \circ \left( w_{i1} x_i + \sum_{j=2}^{k_s-1} w_{ij} x_{\pi_i(j)} + \sum_{j=k_s+1}^{k-k_t} w_{ij} x_{\pi_{i_p}(j)} \right) \tag{3}$$

$$\tilde{y}_i \leftarrow w_{i1} y_i + \sum_{j=2}^{k_s-1} w_{ij} y_{\pi_i(j)} \tag{4}$$

Here, $k_s$ denotes the number of private images, $k_t$ the number of public images, and $\pi_{i_p}$ is a random permutation over the set $\{n+1, \ldots, n+m\}$. Note that public images are used solely as a source of structured noise, and their labels are not included in the mix.

## 3.2 Singularization algorithm

The principal aim of the singularization algorithm is to transform the original dataset $\{(x_i, y_i)\}_{i=1}^n$ into a new set $\{(\tilde{x}_i, \tilde{y}_i)\}_{i=1}^n$ such that the resulting dataset preserves the discriminative characteristics of the original data, while ensuring that the original data cannot be recovered.

A key vulnerability of the InstaHide algorithm arises from the possibility that two encoded samples may share the same private input during the mixup process. This stems from InstaHide's encoding strategy, where each encoded sample is formed as a convex combination of two data points from the original private dataset and $k-2$ additional samples drawn from public sources. The inclusion of multiple private samples in each encoded combination allows an attacker to group encoded samples that share a common private component, making it possible to reconstruct the original private image from these clusters. For the attacker, the repeated appearance of a private sample across different encodings acts as a persistent signal, while the public components serve as noise that can be filtered out through aggregation.

There are three primary distinctions between our approach and InstaHide. First, our method constructs each encoded input using exactly two private data points ($k = 2$), and does not incorporate any public data. This design choice is motivated by security considerations. While InstaHide suggests increasing $k$ to mitigate brute-force attacks on the subset sum problem, Carlini et al. (2021a) has demonstrated that larger values of $k$ can actually reduce security. Specifically, when the mixing weights are known, the resulting linear system becomes easier to solve, making it more vulnerable to attack. By fixing $k = 2$ and relying exclusively on private data, our approach avoids these vulnerabilities.

Second, we introduce noise only to the second private data point before performing the mixup operation. This strategy is intended to tightly couple the noise with the private information, thereby making it significantly more difficult for an adversary to recover the original data. When the noise level is sufficiently high, inverting the process becomes an ill-conditioned problem, which further enhances security.

Finally, our method does not require sign-flipping masks or the use of public images. Previous attacks Carlini et al. (2021a); Chen et al. (2020); Luo et al. (2022) have shown that sign-flipping masks can be circumvented by analyzing the absolute values of mixed images, thus limiting their effectiveness as a privacy mechanism. Our singularization algorithm is presented in Algorithm 1.

---

**Algorithm 1** Singularized Mixup

---

**Require:** Dataset $\{(x_i, y_i)\}_{i=1}^n$; error norm $r$
**Ensure:** Mixed dataset $\{(\tilde{x}_i, \tilde{y}_i)\}_{i=1}^n$
  1: $\pi \sim \text{Uniform}(S_n)$
  2: **for** each $i = 1$ to $n$ **do**
  3:    $w_i \sim \text{Uniform}([0,1]^2)$ and normalize such that $\|w_i\|_1 = 1$ and $\|w_i\|_\infty \leq \alpha$
  4:    $e_i \sim \text{Uniform}(\mathbb{S}(0, r))$
  5:    $\tilde{x}_i \leftarrow w_{i1} x_i + w_{i2}(x_{\pi(i)} + e_i)$
  6:    $\tilde{y}_i \leftarrow w_{i1} y_i + w_{i2} y_{\pi(i)}$
  7: **end for**
  8: **return** $\{(\tilde{x}_i, \tilde{y}_i)\}_{i=1}^n$

---

### 3.3 PRACTICAL INSTANTIATION

Similar to InstaHide Huang et al. (2020), the primary application of our algorithm is in privacy-preserving collaborative training. Suppose there are multiple parties, each possessing a private local dataset. These parties aim to jointly train a deep neural network on the combined data without exposing the sensitive information contained in their individual datasets. The following general framework demonstrates how Algorithm 1 can be integrated to achieve this goal:

1. All parties agree on a common preprocessing technique to be applied locally. For instance, in the context of image data, participants may choose to standardize the images or extract feature representations using a publicly available pretrained model, such as ResNet He et al. (2016).

2. Each party independently transforms its local dataset by applying Algorithm 1, thereby generating a set of mixup samples. Each sample consists of a mixup example and its corresponding mixup label.

3. The resulting data is then transmitted to a central server, which is responsible for training the deep learning model. Upon completion, the trained model is distributed back to the parties for local use.

The security of this protocol depends on the effectiveness of Algorithm 1 in protecting the privacy of local datasets. In particular, the central server, which only receives the mixup samples, should not be able to reconstruct the original data from these representations.

## 4 SECURITY ANALYSIS

In this section, we analyze the security of the proposed scheme and evaluate its resilience to the three principal attacks on the InstaHide framework, as presented in Carlini et al. (2021a), Chen et al. (2020), and Luo et al. (2022). We begin by reviewing the attack strategy of Carlini et al. (2021a), which applies directly to InstaHide without additional assumptions and forms the foundation for subsequent attacks.

The attack of Carlini et al. (2021a) aims to recover the noisy linear system of equations generated by the InstaHide encoding procedure. The adversary's first task is to identify which private images contribute to each encoded sample. To do so, the attacker generates encoded images using publicly available data and trains a neural network to predict whether two encoded images share a common private source image. Although InstaHide introduces random sign flips that might be expected to impede this process, the authors show that taking the absolute value of each pixel effectively removes this obstacle. Once trained, the comparison network allows the adversary to infer the co-occurrence of private images across encoded samples. Since the mixing weights are revealed through the encoded labels, the attacker can then assemble a noisy linear system whose noise arises from the public images included in each mixup. Because the public images vary across samples, this noise behaves approximately as mean-zero Gaussian and averages out across many equations. Solving this system enables the attacker to reconstruct the private images up to a global sign per pixel. A final recoloring step is applied to improve visual fidelity.

In contrast, the attack of Chen et al. (2020) relies on an explicit distributional assumption: the original images are modeled as Gaussian. Under this assumption, the absolute values of the encoded samples follow a folded Gaussian distribution. Given sufficiently many mixup samples, the adversary can estimate their covariance matrix, which corresponds to the Gram matrix of the mixing weight vectors. From this Gram matrix, the adversary can determine which private images participate in each mixup. As in Carlini et al. (2021a), this identification step enables the construction of a linear inverse problem whose solution yields the private images up to pixel-wise sign ambiguities.

To mitigate the attack of Carlini et al. (2021a), Luo et al. (2022) proposes introducing geometric augmentations—such as shifting, cropping, rotation, and translation—to disrupt pixel-wise alignment before mixup. This defense seeks to prevent the formation of a consistent linear system that attackers could invert. Their approach shares a broad motivation with ours: both aim to ensure that identical images do not reappear across multiple encoded samples. However, Luo et al. (2022) demonstrates that such augmentations can be circumvented. Specifically, the attacker trains a comparison network, as in Carlini et al. (2021a), to detect whether two encoded samples share a common private image, even under geometric transformations. Encoded samples containing the same private image are then clustered. Within each cluster, a fusion-denoising pipeline is applied: a convolutional network first downsamples the encoded images to reduce geometric variability, a transpose CNN upsamples the result, and multiple outputs are fused (by averaging or max-pooling) before being passed through a denoising network that reconstructs the private image.

The attacks of Carlini et al. (2021a) and Chen et al. (2020) are ineffective against our method because the injected noise disrupts the formation of stable linear systems. Moreover, unlike Luo et al. (2022), our scheme does not rely on geometric transformations, thereby limiting the applicability of fusion-denoising attacks. Although our mechanism is designed to prevent adversaries from forming clusters or reconstructing underlying linear systems, we adopt a conservative security model in which the adversary is assumed capable of doing so. This assumption is motivated by a common feature of all three attacks: the ability to determine whether two encoded images share a private component. In both Carlini et al. (2021a) and Chen et al. (2020), the random sign mask is rendered ineffective by taking absolute values, while in Luo et al. (2022), geometric transformations do not prevent clustering.

To assess the security of our singularized mixup algorithm, we characterize the reconstruction error faced by an adversary attempting to invert the system of equations induced by Algorithm 1. We assume the attacker has access to both the encoded samples and the mixing weights, and seeks to recover the original data. In Theorem 4.1, we examine an adversary that does not leverage any structural prior, such as assuming the unknowns are images or adhere to particular statistical constraints. The theorem shows that the expected Euclidean recovery error scales linearly with the noise radius $r$.

**Theorem 4.1.** *Let $X \in \mathbb{R}^{n \times d}$ have rows $x_i^\top$. Algorithm 1 produces*

$$\tilde{x}_i = w_{i1} x_i + w_{i2} (x_{\pi(i)} + e_i), \qquad e_i \overset{i.i.d.}{\sim} \mathrm{Uniform}(\mathbb{S}(0, r)),$$

*with $\|w_i\|_1 = 1$ and $\|w_i\|_\infty \leq \alpha$. Let $P_\pi$ be the permutation matrix of $\pi$ and define*

$$D_1 := \mathrm{diag}(w_{11}, \ldots, w_{n1}), \qquad D_2 := \mathrm{diag}(w_{12}, \ldots, w_{n2}), \qquad W := D_1 + D_2 P_\pi,$$

*and assume $W$ is invertible. Then*

$$\tilde{X} = WX + E, \qquad E_i = w_{i2} e_i.$$

*For any estimator $\hat{x}_i = \hat{x}_i(\tilde{X}, W)$,*

$$\sup_{X \in \mathbb{R}^{n \times d}} \mathbb{E}\big[\|x_i - \hat{x}_i(\tilde{X}, W)\|_2^2\big] \geq r^2 T_i,$$

*where*

$$T_i := \sum_{\ell=1}^n (W^{-1})_{i\ell}^2 \, w_{\ell 2}^2.$$

*Proof.* Since $W$ is invertible, define

$$Y := W^{-1}\tilde{X} = X + Z, \qquad Z := W^{-1}E.$$

Because $E_\ell = w_{\ell 2} e_\ell$ and $e_\ell$ are independent, mean-zero, and isotropic with

$$\text{Cov}(e_\ell) = \frac{r^2}{d} I_d,$$

we have

$$\text{Cov}(E_\ell) = w_{\ell 2}^2 \frac{r^2}{d} I_d.$$

Thus

$$\text{Cov}(Z_i) = \sum_{\ell=1}^{n} (W^{-1})_{i\ell}^2 \, \text{Cov}(E_\ell) = \frac{r^2}{d} \, T_i \, I_d,$$

and hence

$$\mathbb{E}\|Z_i\|_2^2 = \text{tr}(\text{Cov}(Z_i)) = r^2 \, T_i.$$

Fix any estimator $\hat{x}_i(\tilde{X}, W)$ and write $\delta(Y) := \hat{x}_i(\tilde{X}, W)$. Consider matrices $X$ with all rows equal to zero except the $i$th. For such $X$,

$$Y_i = x_i + Z_i, \qquad (Y_{-i}, Z_i) \text{ independent of } x_i.$$

Let $m(Y_i) := \mathbb{E}[\delta(Y) \mid Y_i]$. By Jensen's inequality,

$$\mathbb{E}\big[\|x_i - m(Y_i)\|_2^2\big] \;\leq\; \mathbb{E}\big[\|x_i - \delta(Y)\|_2^2\big] \qquad \text{for every } x_i \in \mathbb{R}^d.$$

Now restrict attention to this one-dimensional family $X$ parametrized by $x_i$. We obtain the $d$-dimensional location model

$$Y_i = x_i + Z_i, \qquad x_i \in \mathbb{R}^d,$$

with fixed noise $Z_i$. A standard Bayesian lower bound for location models implies

$$\sup_{x_i \in \mathbb{R}^d} \mathbb{E}\big[\|x_i - m(Y_i)\|_2^2\big] \;\geq\; \mathbb{E}\|Z_i\|_2^2.$$

Combining the inequalities,

$$\sup_{X} \mathbb{E}\big[\|x_i - \hat{x}_i(\tilde{X}, W)\|_2^2\big] \;\geq\; \mathbb{E}\|Z_i\|_2^2 = r^2 \, T_i.$$

This completes the proof. $\qquad\qquad\qquad\qquad\qquad\qquad\qquad\qquad\qquad\qquad\qquad\qquad$ $\square$

While Theorem 4.1 provides a tight minimax lower bound on the MSE for estimators that lack prior knowledge, it does not prescribe how to choose the noise norm $r$. Our goal is to select $r$ such that encoded samples do not leak meaningful information about the originals. We formalize this through the signal-to-noise ratio (SNR) associated with the decomposition of each encoded sample into signal and interference components. Motivated by the defense rationale in Luo et al. (2022), where distinct augmented images are used to prevent redundancy in mixup inputs, we choose $r$ proportional to the typical separation between data points. Theorem 4.2 specifies how to select the scaling parameter $mf$ and thereby set $r$ as a multiple of the average inter-sample distance so that the resulting SNR falls below a prescribed threshold $\tau$.

**Theorem 4.2.** *Let $\{x_i\}_{i=1}^{n} \subset \mathbb{R}^d$ be i.i.d. samples of a subgaussian random vector $X$ with $\mathbb{E}[X] = 0$ and covariance $\Sigma$. Define*

$$V \;=\; \mathbb{E}\big[\|X\|_2^2\big] \;=\; \text{tr}(\Sigma), \qquad D \;=\; \mathbb{E}\big[\|X - X'\|_2\big],$$

*for an independent copy $X'$. Set*

$$c \;:=\; \frac{D^2}{2V} \;\in\; (0, 1],$$

*so that $D^2 = 2cV$ (Jensen and independence give $D^2 \leq 2V$). Algorithm 1 outputs, for each $i$,*

$$\tilde{x}_i \;=\; w_{i1} x_i \;+\; (1 - w_{i1})\big(x_{\pi(i)} + e_i\big),$$

*where $0 \leq w_{i1} \leq \alpha$ almost surely for a given $\alpha \in (0, 1)$, and the error vectors satisfy*

$$\|e_i\|_2 \;=\; r, \qquad r \;=\; m_f \, D.$$

*Assume that $w_{i1}$ is independent of the data and that $e_i$ is independent of the data and of $w_{i1}$. Define the signal and interference components*

$$S_i = w_{i1} x_i, \qquad I_i = (1 - w_{i1})(x_{\pi(i)} + e_i),$$

*and the expected signal-to-noise ratio*

$$\mathrm{SNR} = \frac{\mathbb{E}\|S_i\|_2^2}{\mathbb{E}\|I_i\|_2^2}.$$

*Then*

$$\mathrm{SNR} \leq \frac{\alpha^2}{(1-\alpha)^2\left(1+\frac{r^2}{V}\right)} = \frac{\alpha^2}{(1-\alpha)^2\left(1+\frac{m_f^2 D^2}{V}\right)} = \frac{\alpha^2}{(1-\alpha)^2\left(1+2c\,m_f^2\right)}.$$

*Consequently, for any target $\tau > 0$,*

$$\mathrm{SNR} \leq \tau \quad \text{whenever} \quad m_f \geq \sqrt{\frac{1}{2c}\left(\frac{\alpha^2}{\tau(1-\alpha)^2} - 1\right)}.$$

*Proof.* Since $0 \leq w_{i1} \leq \alpha$ a.s. and $w_{i1}$ is independent of $x_i$,

$$\mathbb{E}\|S_i\|_2^2 = \mathbb{E}\big[w_{i1}^2\|x_i\|_2^2\big] = \mathbb{E}[w_{i1}^2]\,\mathbb{E}\big[\|X\|_2^2\big] \leq \alpha^2 V.$$

For the interference, using independence of $w_{i1}, X', e_i$, zero-mean $\mathbb{E}[X'] = 0$, and the polarization identity,

$$\mathbb{E}\|I_i\|_2^2 = \mathbb{E}\big[(1-w_{i1})^2\|X' + e_i\|_2^2\big]$$
$$= \mathbb{E}\big[(1-w_{i1})^2\big]\,\mathbb{E}\big[\|X'\|_2^2\big] + \mathbb{E}\big[(1-w_{i1})^2\big]\,\mathbb{E}\big[\|e_i\|_2^2\big] + 2\,\mathbb{E}\big[(1-w_{i1})^2\big]\,\mathbb{E}[\langle X', e_i\rangle].$$

The cross term vanishes since $\mathbb{E}[X'] = 0$ and $e_i$ is independent of $X'$. Hence

$$\mathbb{E}\|I_i\|_2^2 = \mathbb{E}\big[(1-w_{i1})^2\big]\big(V + \mathbb{E}\|e_i\|_2^2\big).$$

Because $0 \leq w_{i1} \leq \alpha$ a.s., we have $(1 - w_{i1}) \geq (1 - \alpha)$ a.s., whence

$$\mathbb{E}\big[(1-w_{i1})^2\big] \geq (1-\alpha)^2.$$

By construction $\|e_i\|_2 = r$ deterministically, so $\mathbb{E}\|e_i\|_2^2 = r^2$. Therefore,

$$\mathbb{E}\|I_i\|_2^2 \geq (1-\alpha)^2(V + r^2).$$

Taking the ratio gives

$$\mathrm{SNR} = \frac{\mathbb{E}\|S_i\|_2^2}{\mathbb{E}\|I_i\|_2^2} \leq \frac{\alpha^2 V}{(1-\alpha)^2(V + r^2)} = \frac{\alpha^2}{(1-\alpha)^2\left(1+\frac{r^2}{V}\right)}.$$

Substituting $r = m_f D$ yields

$$\mathrm{SNR} \leq \frac{\alpha^2}{(1-\alpha)^2\left(1+\frac{m_f^2 D^2}{V}\right)} = \frac{\alpha^2}{(1-\alpha)^2\left(1+2c\,m_f^2\right)},$$

because $D^2 = 2cV$ by definition of $c$. Finally, solving

$$\frac{\alpha^2}{(1-\alpha)^2\left(1+2c\,m_f^2\right)} \leq \tau$$

for $m_f$ gives

$$m_f \geq \sqrt{\frac{1}{2c}\left(\frac{\alpha^2}{\tau(1-\alpha)^2} - 1\right)},$$

which completes the proof. $\square$

Finally, although Theorem 4.1 characterizes the fundamental difficulty of inversion for adversaries without prior information, practical attackers may possess substantial knowledge about the structure or distribution of the underlying data. To bridge this gap, our experimental evaluation assesses the performance of both linear and nonlinear estimators in realistic scenarios, thereby providing a more comprehensive understanding of the scheme's robustness against adversaries capable of exploiting informative priors.

## 5 EXPERIMENTS

Our experiments are designed to evaluate both the accuracy loss relative to InstaHide and the attack resilience of our proposed algorithm.

**Setup**: We conduct our experiments on three widely used benchmark datasets: MNIST LeCun (1998), CIFAR-10, CIFAR-100 Krizhevsky et al. (2009) and Tiny-ImageNet Le & Yang (2015). All implementations are carried out using the PyTorch framework Paszke et al. (2019). For the classification accuracy experiments, we utilize feature representations obtained from the output of the final convolutional layer of publicly available pretrained image models. Specifically, feature maps are extracted using a pretrained ResNet-18 model for MNIST and CIFAR-10, and a pretrained ResNet-50 model for CIFAR-100 and Tiny-ImageNet.

**Security**:

In the following, we examine two distinct attack strategies for recovering the underlying images from their mixed representations. The first is a linear inversion attack, which exploits the known linear mixing process to directly reconstruct the sources. The second is a non-linear reconstruction attack based on U-Net architectures, allowing the adversary to learn a more flexible, data-driven inverse mapping. We evaluate the quality of the recovered images using the SNR for a range of values of $\tau$, with the noise norm chosen according to Theorem 4.2. In particular, we estimate the average distance between two randomly selected images directly from the data and set the noise level $r$ to be this empirical average multiplied by the scaling factor prescribed by the theorem. This procedure ensures that the noise magnitude is consistent with the theoretical regime under consideration.

For the linear inversion attack, we follow a gradient-descent–based reconstruction procedure inspired by Luo et al. (2022). In this setting, the adversary is assumed to know the linear mixing matrix and uses this information to guide the recovery process. The loss function is built around a linear reconstruction term that measures the mean squared error between the observed mixtures and the linearly recomposed images obtained by applying the known mixing weights to the current estimates. To stabilize this inversion, the attacker incorporates generic priors on natural images through additional regularization terms: a total-variation penalty that promotes spatial smoothness and $L_2$ penalty that discourages unrealistically large pixel values. These regularizers encode broad assumptions about natural images—namely smoothness and bounded intensity—without introducing any nonlinear modeling. Optimization is carried out with Adam, and after each update the recovered images are clipped to a fixed range to maintain plausible pixel values. The resulting reconstruction performance is reported in Table 1, and Figure 3 shows the best recovered image for CIFAR-10 under the linear attack. Results for all datasets are provided in Appendix G.

Table 1: Average reconstruction SNR (dB) $\pm$ standard deviation for the linear attack for different noise levels $\tau$.

| $\tau$ | MNIST | CIFAR-10 | CIFAR-100 | Tiny-ImageNet |
|---|---|---|---|---|
| $10^0$ | $8.19\pm0.31$ | $0.85\pm2.69$ | $0.94\pm3.01$ | $1.24\pm2.47$ |
| $10^{-1}$ | $0.09\pm0.14$ | $-5.06\pm2.29$ | $-4.86\pm2.52$ | $-4.60\pm2.01$ |
| $10^{-2}$ | $-2.79\pm0.09$ | $-6.92\pm2.10$ | $-6.66\pm2.30$ | $-6.42\pm1.79$ |
| $10^{-3}$ | $-3.65\pm0.08$ | $-7.45\pm2.05$ | $-7.17\pm2.24$ | $-6.94\pm1.75$ |
| $10^{-4}$ | $-3.90\pm0.09$ | $-7.61\pm2.03$ | $-7.32\pm2.22$ | $-7.09\pm1.73$ |
| $10^{-5}$ | $-3.98\pm0.09$ | $-7.66\pm2.03$ | $-7.37\pm2.22$ | $-7.14\pm1.73$ |
| $10^{-6}$ | $-4.01\pm0.09$ | $-7.67\pm2.03$ | $-7.38\pm2.22$ | $-7.15\pm1.73$ |

For the nonlinear attack, we replace the analytic inversion used in the linear setting with a learned reconstruction model based on a U-Net architecture. The adversary is assumed to have access to a collection of public images, which are used to train the network to map mixed inputs back to clean sources. In our experiments, Tiny-ImageNet serves as the public dataset for training, while CIFAR-10 plays the role of the private dataset used for evaluation. A separate U-Net is trained for each value of $\tau$, ensuring that the attacker can adapt to the corresponding noise magnitude. Training is performed using an $\ell_1$ reconstruction loss together with total-variation regularization, and predictions are kept within valid intensity ranges through standard normalization and clamping. The resulting reconstruction SNR for all $\tau$ values are reported in Table 2. Figure 2 shows the best recovered CIFAR-10 example across noise levels.

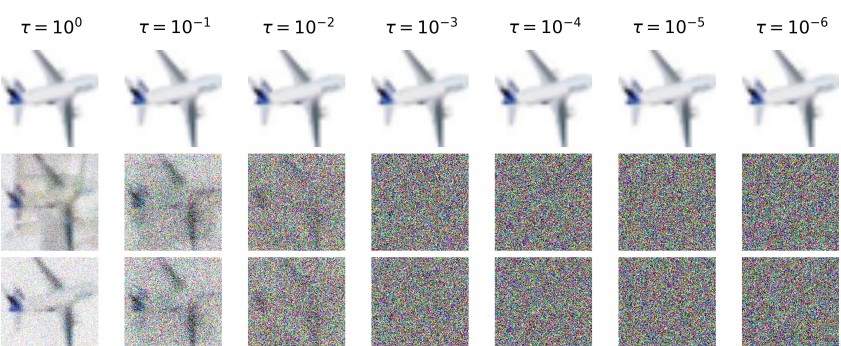

Figure 1: Recovered images under the linear attack for decreasing noise levels $\tau$

Table 2: Average reconstruction SNR (dB) $\pm$ standard deviation for the nonlinear attack across noise levels $\tau$.

| $\tau$ | $10^0$ | $10^{-1}$ | $10^{-2}$ | $10^{-3}$ | $10^{-4}$ | $10^{-5}$ | $10^{-6}$ |
|---|---|---|---|---|---|---|---|
| | $7.76\pm3.46$ | $7.27\pm3.39$ | $5.44\pm2.93$ | $2.84\pm2.28$ | $-0.13\pm0.54$ | $-0.13\pm0.53$ | $-0.14\pm0.58$ |

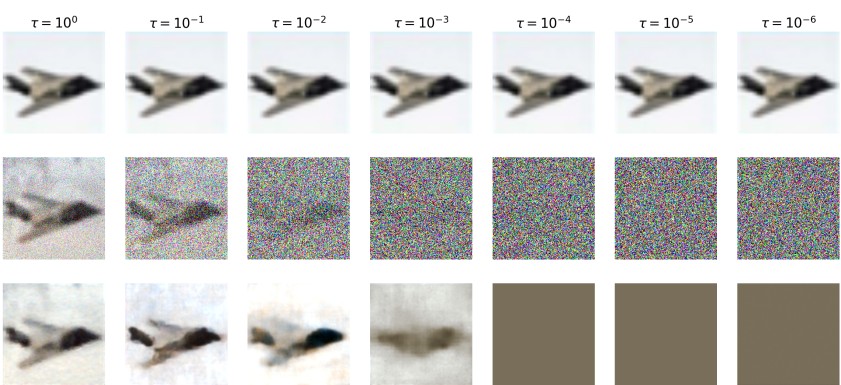

Figure 2: Nonlinear (U-Net) attack on CIFAR-10: ground truth, mixed inputs, and recovered images across noise levels $\tau$

**Accuracy**:

The security analysis indicates that, although the nonlinear attack is noticeably more powerful than the linear one for $\tau \geq 10^{-4}$, the original image remains unrecoverable across all datasets even at this relatively small noise level. Since $\tau = 10^{-4}$ already suffices to prevent meaningful reconstruction, we adopt a conservative stance and conduct all accuracy experiments at an even stricter privacy setting, using $\tau = 10^{-6}$ uniformly across datasets. At this noise level, we train a standard convolutional classifier on mixed representations and report the resulting test accuracy for each dataset. We then compare these results with the best-performing configuration of InstaHide (with $k = 4$) to quantify the utility–privacy trade-off under a conservative noise regime. The full accuracy results are summarized in Table 3.

Table 3: Test accuracy (%) at $\tau = 10^{-6}$ compared with the best reported InstaHide configuration ($k = 4$).

| Method | MNIST | CIFAR-10 | CIFAR-100 | Tiny-ImageNet |
|---|---|---|---|---|
| InstaHide ($k = 4$) | 99.66 | 91.20 | 74.01 | – |
| Ours ($\tau = 10^{-6}$) | 99.32 | 90.51 | 75.99 | 72.50 |

The accuracy results in Table 3 show that, even under the conservative noise setting $\tau = 10^{-6}$—significantly stricter than what is needed to prevent both linear and nonlinear reconstruction—the loss in predictive performance remains negligible across all datasets. On MNIST and

CIFAR-10, our approach matches the accuracy of InstaHide with $k = 4$, and on CIFAR-100 and Tiny-ImageNet it even yields modest improvements. These findings indicate that strong reconstruction resistance does not come at the expense of meaningful degradation in downstream utility: despite enforcing a noise level far below the threshold where the nonlinear attacker fails (i.e., $\tau \approx 10^{-4}$), the mixed representations still support high classification accuracy. Overall, the results demonstrate that our method provides robust security guarantees while preserving competitive model performance across diverse datasets.

We additionally evaluate our method in a federated learning setting by partitioning the CIFAR-10 dataset across multiple parties. Each party holds a disjoint local subset of the data. We compare (i) a baseline model trained solely on the local subset of Party 0, and (ii) a model trained on the union of all mixed datasets produced by Algorithm 1 across the participating parties. The results in Table 4 show that training on the aggregated mixed representations consistently achieves higher accuracy as the number of parties increases.

Table 4: Federated CIFAR-10 test accuracy (%). The baseline model is trained only on Party 0's local data, while the second model is trained on the union of mixed datasets produced by all parties using Algorithm 1.

| Number of Parties | Baseline (Local Only) | Mixup-Union (All Parties) |
| --- | --- | --- |
| 10 | 87.25 | 90.63 |
| 20 | 85.56 | 90.32 |
| 30 | 84.08 | 88.23 |

## 6 CONCLUSIONS

In this paper, we introduced a singularized mixup mechanism that mixes only two private images at a time while injecting structured noise into the non-target component. This design directly addresses the key vulnerability exploited in attacks on InstaHide, where repeated use of the same private image across many mixtures enables clustering and subsequent reconstruction. By corrupting all but one component in each mixture, our method prevents such clustering and limits what an adversary can infer from any single mixed sample.

We analyzed security under a conservative threat model in which the attacker has full knowledge of the mixing weights and evaluated both a linear inversion attack and a more powerful nonlinear U-Net–based attack. Our experiments show that once the noise magnitude exceeds a modest threshold, neither attack can recover the original image, even when given maximal information about the mixing process. At the same time, using a conservative setting of $\tau = 10^{-6}$—well below the level at which both attacks already fail—the mixed representations retain high utility, with only negligible accuracy loss compared to InstaHide's best $k = 4$ configuration.

While the method is tailored to image data, the singularization principle may extend to other modalities with appropriate noise models. Exploring such extensions presents an interesting direction for future work toward broadly applicable, attack-resistant data mixing schemes.

### REPRODUCIBILITY STATEMENT

All source code used in this study is publicly available, along with detailed instructions to reproduce the experiments described in this paper. The data utilized comes from publicly accessible benchmark datasets. For additional details regarding the experimental setup and procedures, please refer to the Appendix E.

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

## A  CHEN ATTACK

The attack described in Chen et al. (2020) simplifies the InstaHide problem by assuming that the matrix $X \in \mathbb{R}^{d \times n}$ of images is Gaussian, i.e., its entries are chosen i.i.d. from $\mathcal{N}(0, 1)$. Let $p_1, \ldots, p_d \in \mathbb{R}^n$ be the rows of $X$. Consider $w_{i1}, dots, w_m \in \mathbb{R}^n$ the unknown selection vectors chosen from a distribution $\mathcal{D}$. $S \subset \{1, \ldots, m\}$ be the coordinates of the public imges and $S^c = \{1, \ldots, n\}$ $S$ be the coordinates of the private images. Let $[v]_S \in \mathbb{R}^{|S|}$ be the restriction of a vector $v$ to the coordinates indexed by $S$. Each selection sector generates an encoded image as:

$$\tilde{x}_i = |X w_i| \tag{5}$$

In InstaHide, the sign of each pixel from an encoded image is randomly flipped, but as the authors remark, the two notations are interchangeable.

The attack goes like follows:

1. **Learning the public coordinates of any selection vector** In the first step, the attacker determines the weights associated with the public images from each selection vector. Considering the matrix
$$N = \mathbb{E}_{p, \tilde{x}} \left[ \tilde{x}^2 \cdot ([p]_S [p]_S^\intercal - \text{Id}) \right]$$
where $\tilde{x} = |\langle w, p \rangle|$, $p \sim \mathcal{N}(0, \text{Id})$. It can be proven that $N$ is a rank-1 matrix proportional to $[w]_S [w]_S^\intercal$. Moreover, $N$ can be approximated by

$$\hat{N} = \frac{1}{d} \sum_{i=1}^{d} \tilde{x}_i^2 \cdot ([p_i]_S [p_i]_S^\intercal - \text{Id}).$$

2. **Recovering the Gram Matrix** Since the previous steps recover the coordinates of the public images in each selection vector, for the simplicty of the description, consider that all images are private, i.e., $S^c = \{1, \ldots, n\}$. Consider the matrix $\tilde{X} \in \mathbb{R}^{m \times d}$ where each line is an encoded images:

$$\tilde{X} = \begin{pmatrix} |\langle p_1, w_1 \rangle| & \cdots & |\langle p_d, w_1 \rangle| \\ \vdots & \ddots & \vdots \\ |\langle p_1, w_m \rangle| & \cdots & |\langle p_d, w_m \rangle| \end{pmatrix} \tag{6}$$

We can use the columns of $\tilde{X}$ to estimate the covariance matrix $\tilde{M}$ of the folded Gaussian distribution $\mathcal{N}^{\text{fold}}(0, M)$, since each column is drawn independently from this distribution. The covariance matrix $M$ is, in fact, the rescaled $m \times m$ Gram matrix whose entries are proportional to the dot product of any two selection vectors; that is, the element at position $(i, j)$ in the matrix $M$ is given by $k \cdot \langle w_i, w_{ij} \rangle$.

3. **Floral submatrices**

   The previous step of the attack shows the dot product between any two selection vectors, i.e., $\langle w_i, w_{ij} \rangle$, thus the attacker knows how many private images are common between two encoded images. In order to identify which private images are common (not only how many), the attacker identifies in $M$ floral submatrices. The rows/columns of a floral submatrix can be indexed by all subsets of size $k$ of a set of $k+2$ elements where its entries are the intersection sizes between the subsets. More intuitively, the attacker exploits the fact that the subsets of size $k$ of of the set $\{1, \ldots, k+2\}$ are uniquely identified by their pairwise intersection sizes.

4. **Determining the private images**

   Suppose the attacker has identified a floral matrix in the previous steps, which corresponds to the selection vectors $w_{i1}, \ldots, w_{i_t}$, where $t = \binom{k+2}{k}$. The structure of the floral matrix encodes information about the indices of private images that are common between pairs of selection vectors. Specifically, the row and column indices of the matrix indicate which private images are shared. This allows the attacker to construct a system of equations of the form $|\langle w_{ij}, p_l \rangle| = \tilde{x}_l$ for all $l \in 1, \ldots, d$, where $p_l$ denotes the private images and $\tilde{x}_l$ are known quantities.

   From another perspective, each row or column of the floral submatrix can be indexed by a subset of size $k$ from a set of size $k+2$. Each element in such a subset represents the index of a private image. For any given element in the floral matrix—which itself is a submatrix of the Gram matrix $M$—the position of the element along the rows provides the attacker with a set of $k$ private image indices, while the position along the columns provides another set of $k$ indices. By intersecting these two sets, the attacker can determine which private images are common between the selection vectors associated with the corresponding row and column. Solving the resulting system of equations enables the attacker to recover the indices of the private images.

# B LUO ATTACK

In Luo et al. (2022), the authors observed that the method proposed by Carlini in Carlini et al. (2021a) can be mitigated by applying data augmentation before the mixup process. To address this, they introduce a new approach that successfully bypasses this mitigation strategy. Their method operates as follows:

1. In the first step, the attacker computes the absolute value of each pixel in every encoded image.

2. Next, a similarity score is calculated for every pair of encoded images to determine, with high probability, whether a given pair is derived from the same private image. To compute this score, the authors propose a comparative network that takes as input both high-resolution and low-resolution versions of the image pairs. This approach yields better results than the standard ResNet architecture used by Carlini. Based on the similarity scores,

the attacker clusters the encoded images, with each cluster corresponding to a distinct private image.

3. For each cluster obtained in the previous step, the attacker re-weights all encoded images using the weights associated with the corresponding private image. These weights can be easily inferred from the associated encoded labels. Subsequently, a neural network is trained to perform image relaxation and fusion. This strategy counteracts the effects of geometric image augmentation by generating a set of features that are invariant to geometric transformations. An initial version of the private image is then constructed in the fusion step by combining these feature maps.

4. In the final step, the attacker trains an additional neural network to denoise the image produced in the previous stage.

## C  CARLINI ATTACK

The attack consists of two main stages. In the first stage, the attacker determines the two private images used to generate each encoded image during the mixing process. In the second stage, the attacker reconstructs the private images by solving a noisy linear system of equations:

1. The attacker computes the absolute value of each mixup encoding to counteract the random sign changes introduced by the mask $\sigma_i$ in (3).

2. To identify whether two encoded images share at least one common private image, the attacker calculates a similarity function between each pair of encoded images. This similarity function is approximated using a neural network trained on public data transformed via the mixup algorithm. Using the similarity scores, the attacker constructs a weighted graph where vertices represent encoded samples, and edge weights correspond to the similarity function's output.

3. Based on the weighted graph, the attacker identifies densely connected cliques, enabling clustering of encoded samples that share a common private image. Each cluster is represented as a set $S_i$, $1 \leq i \leq n$, where each set contains encoded samples derived from the same private image.

4. Since each encoded image is generated by mixing two private images, the attacker constructs a bipartite similarity graph connecting encoded images to the sets identified in the previous step. Edge weights represent the distance between an encoded image $x_i$ and a set $S_i$. This step determines, for each encoded image, the two sets corresponding to the private images used in its construction.

5. Using the bipartite graph, the attacker maps each encoded image to two sets, representing the private images involved in its generation during the mixup process.

6. The attacker recovers the weights used to generate each encoded image by analyzing the mixup of the labels, as described in (4). Since the labels are one-hot encoded, recovering the associated weights is straightforward.

7. Finally, the attacker constructs a matrix $B \in \mathbb{R}^{n \times d}$, where each row corresponds to an encoded image $\tilde{x}_i$, i.e., $B_i = \tilde{x}_i$. A sparse matrix $M \in \mathbb{R}^{n \times n}$ is also constructed, where each row contains two non-zero entries representing the weights $w^i 1$ and $w_2^i$ associated with the private images used to compute the corresponding encoded image. Let $A \in \mathbb{R}^{n \times d}$ represent the matrix of private images, where each row $A_i = x_i$. The attacker solves the noisy linear system $B = M \cdot A + e$, where $e$ represents the public images used in the mixup. This system can be efficiently solved using gradient descent.

## D  MORE RELATED WORK

Liu et al. (2020) proposed a different approach, where a classifier is trained on mixup samples and images to produce mixup results that can later be de-mixed. Unlike previous methods, this approach does not involve training on mixup samples followed by inference on original data. Instead, both training and inference are performed on mixup data, with the inference process generating mixup results that can then be used to recover the correct labels.

In a more recent study, Wang et al. (2024) proposed a mixup-like approach to mitigate model inversion attacks on face recognition systems. Instead of mixing images directly, the authors suggested mixing samples in the frequency domain. Additionally, they employed a reinforcement learning strategy to dynamically determine the number of images to mix, balancing privacy and utility. Similarly, Xiang et al. (2023) introduced a mixing strategy to preserve image privacy during training. Their method involves splitting each image into multiple blocks and replacing parts of these blocks with corresponding blocks from other images with the same label. In another study, Li et al. proposed a new privacy metric called Visual Feature Entropy (VFE), calculated for a region of an image as the sum of squared gradients with respect to both axes. This metric aims to quantify the amount of information that needs protection by analyzing the entropy of a region. The authors' mixing strategy involves shuffling pixels within an image based on the VFE metric. Although this method does not involve computing a weighted sum, it can be interpreted as a form of intra-image data mixing. Eloul et al. (2024) present the concept of mixing gradients in federated learning to enhance security against gradient inversion attacks. Although their method does not involve using random weights for gradient mixing, their straightforward approach of directly averaging gradients across a batch, combined with modifications to the loss function, significantly improves resistance to gradient inversion attacks.

The concept of data mixing is rooted in the broader idea of learnable obfuscation, which encompasses techniques designed to transform data in a way that allows algorithms to learn from the transformed data while safeguarding the privacy of the original data He et al. (2020); Yala et al.; Taki & Mastorakis (2024); Popescu et al. (2022); Nythia et al. (2017). For instance, in Nythia et al. (2017), the authors propose using the Arnold transformation to scramble images before inputting them into a face recognition system. This transformation rearranges image pixels by mapping each pixel to a new location determined by a linear transformation.

In Popescu et al. (2022), a method combining Variational Autoencoders (VAEs) with a substitution technique is introduced to protect medical images during neural network analysis. The approach involves training a VAE to reconstruct the image and then applying a substitution table to the latent space representation of the data. Similarly, Taki & Mastorakis (2024) presents a method to ensure the privacy of both training data and neural network architecture. For image data, the authors propose transforming it into a higher-dimensional space. To protect the architecture, they introduce random subnetworks with synthetic parameters that do not affect the network's accuracy or data flow.

The NeuraCrypt method, proposed in Yala et al., protects data privacy by transforming it with a random neural network. This approach is extended to enable privacy-preserving collaborative training, where all parties share transformed data with a central server. For the server to learn patterns from the combined datasets, all parties must use the same neural network for data transformation. Finally, He et al. (2020) introduces a privacy-preserving method that applies a linear transformation to each data sample. The authors also provide formal proofs demonstrating the information-theoretic security of their approach under specific conditions.

A common characteristic of learnable obfuscation techniques is that the same transformation—though potentially generated using independently chosen random parameters—must be applied to all samples in the dataset being protected. This creates a notable vulnerability: such techniques cannot provide security against chosen-plaintext attacks. This limitation, formally introduced and proven in Xiao et al. (2024), highlights an inherent weakness in these methods. Informally, learnable obfuscation can protect the privacy of plaintext data only under the assumption that the attacker does not have prior knowledge of the original data.

At first glance, this assumption may seem reasonable, as protecting data already known to an attacker might appear unnecessary. However, in practical scenarios, this assumption often fails. For instance, to improve the generalization capabilities of a machine learning model, private datasets are frequently augmented with publicly available data. For example, a private image dataset might be enriched with images from CIFAR-100 Krizhevsky et al. (2009). To preserve the discriminative properties of the data and enable the model to generalize, the added public data must undergo the same transformation used to protect the private dataset.

This practice introduces a significant risk: an attacker with access to both the original public dataset and its transformed version could potentially design algorithms to reverse-engineer the transformation applied to the private data. An example of this vulnerability is described in Carlini et al.

(2021b), where the authors successfully developed an algorithm to solve the NeuraCrypt challenge Yala et al., effectively bypassing the intended privacy protections.

# E    EXPERIMENTAL DETAILS

All experiments were developed using the PyTorch framework and performed on an NVIDIA L4 GPU with 24 GB of available VRAM. Across all datasets, we consistently used a batch size of 128. For optimization, we employed the `AdamW` optimizer with a weight decay of $1 \times 10^{-4}$. The initial learning rate was set to $0.001$ for all benchmarks, and we utilized a cosine annealing learning rate scheduler.

Our experimental evaluation was conducted on three distinct benchmarks using ResNet architectures (He et al., 2016), with specific configurations detailed in Table 5. A key aspect of our methodology is the exclusive use of the feature extraction layers from these architectures; the final classifier layers were omitted as our focus is on training features. For all datasets we apply similar transformations, which consist of a resize operation (224 pixels height and width) and a normalization. The MNIST dataset is also adjusted such that it has three channels, making it compatible with the chosen architectures.

Table 5: The configurations used within experiments for each dataset. The mean and standard deviation values for MNIST are for a single channel, while for CIFAR datasets they correspond to the (R, G, B) channels.

| Dataset | Architecture | Epochs | Mean | Std. |
|---------|-------------|--------|------|------|
| MNIST | ResNet18 | 120 | 0.1307 | 0.3081 |
| CIFAR-10 | ResNet18 | 200 | [0.4914, 0.4822, 0.4465] | [0.2470, 0.2435, 0.2616] |
| CIFAR-100 | ResNet50 | 200 | [0.5071, 0.4867, 0.4408] | [0.2675, 0.2565, 0.2761] |
| Tiny-ImageNet | ResNet50 | 200 | [0.5071, 0.4867, 0.4408] | [0.2675, 0.2565, 0.2761] |

The training objective was to minimize the following loss function, which is designed for our Mixup implementation:

$$\mathcal{L} = -\frac{1}{N} \sum_{i=1}^{N} \sum_{j=1}^{C} y_{ij} \log(\text{softmax}(\mathbf{o}_i)_j) \tag{7}$$

where $N$ is the batch size, $C$ is the number of classes, $\mathbf{y}_i$ is the (potentially soft) label for sample $i$, and $\mathbf{o}_i$ is the model output for sample $i$. In PyTorch, this is implemented as:

```
loss = -(labels * torch.log_softmax(outputs, dim=1)).sum(dim=1).mean()
```

For the final classification step, we used a custom feed-forward neural network with three dense layers. The first and second of these dense layers are followed by batch normalization, GELU activation, and then a dropout. The precise structure of this neural network is described by the following equation:

$$\begin{aligned}
\texttt{Classifier}(x; n_{\text{cls}}) = \text{Flatten}(x) &\rightarrow \text{Linear}_{in;1024} \rightarrow \text{BN}_{1024} \rightarrow \text{GELU} \rightarrow \text{Dropout}(0.5) \\
&\rightarrow \text{Linear}_{1024;512} \rightarrow \text{BN}_{512} \rightarrow \text{GELU} \rightarrow \text{Dropout}(0.5) \\
&\rightarrow \text{Linear}_{512;n_{\text{cls}}}
\end{aligned} \tag{8}$$

The $n_{\text{cls}}$ term represents the number of classes that the classification must be made on (e.g., 10 for MNIST). The $in$ dimension of the flattened tensor $x$ within the first linear layer is 25088 ($512 \times 7 \times 7$ for ResNet18 and ResNet34).

# F    ATTACK IMPLEMENTATION DETAILS

All attacks use the same preprocessing described in Sec. E, including resizing inputs to $224 \times 224$, ImageNet-style normalization, and converting MNIST to three channels. Experiments are executed on a single NVIDIA L4 GPU (24 GB) with fixed random seed (0).

For the *linear reconstruction attack*, we optimize a recovery tensor using the PyTorch Adam optimizer for 200 steps with learning rate 0.05. We include anisotropic total variation regularization and an $\ell_2$ penalty with default weights $\lambda_{\text{tv}} = 10^{-3}$ and $\lambda_2 = 10^{-4}$. After each step, recovered tensors are clamped back into the normalized image range. Recovery quality is measured using per-image SNR in dB, reported as mean $\pm$ std. A single visualization panel is produced: columns correspond to different $\tau$ values and, for each dataset, three aligned rows display the original, the Mixup input, and the recovered image. The displayed index per dataset is chosen as the best-recovered example at the largest $\tau$.

For the *non-linear attack*, we train a U-Net denoiser on Tiny-ImageNet Mixup pairs and evaluate zero-shot on CIFAR-10. The U-Net follows a three-level encoder–decoder design with base width $B = 48$. The encoder consists of successive DoubleConv blocks with channel progression ($3{\rightarrow}B{\rightarrow}2B{\rightarrow}4B{\rightarrow}8B$), separated by $2 \times 2$ max-pooling. The decoder mirrors this structure with transposed convolutions for upsampling, skip connections from encoder features, DoubleConv blocks with channel progression ($8B{\rightarrow}4B{\rightarrow}2B{\rightarrow}B$), and a final $1{\times}1$ convolution mapping to three output channels. All layers use ReLU activations, and outputs are clamped to the normalized range. Training uses Adam with batch size 32, learning rate $10^{-3}$, 30 epochs, and an $\ell_1$ loss augmented with a small TV penalty ($10^{-4}$). Partner selection is deterministic per index so the same samples align across $\tau$ values. Evaluation again reports SNR (mean $\pm$ std). A single summary panel is generated: the first row repeats the ground-truth image associated with the best reconstruction at the largest $\tau$, while the second and third rows show the corresponding Mixup and recovered images for each $\tau$ (displayed as columns). Only the $\tau$ labels appear above columns.

All runs save a single figure per attack configuration along with a timestamped log containing the full console output.

## G    FULL FIGURE 3

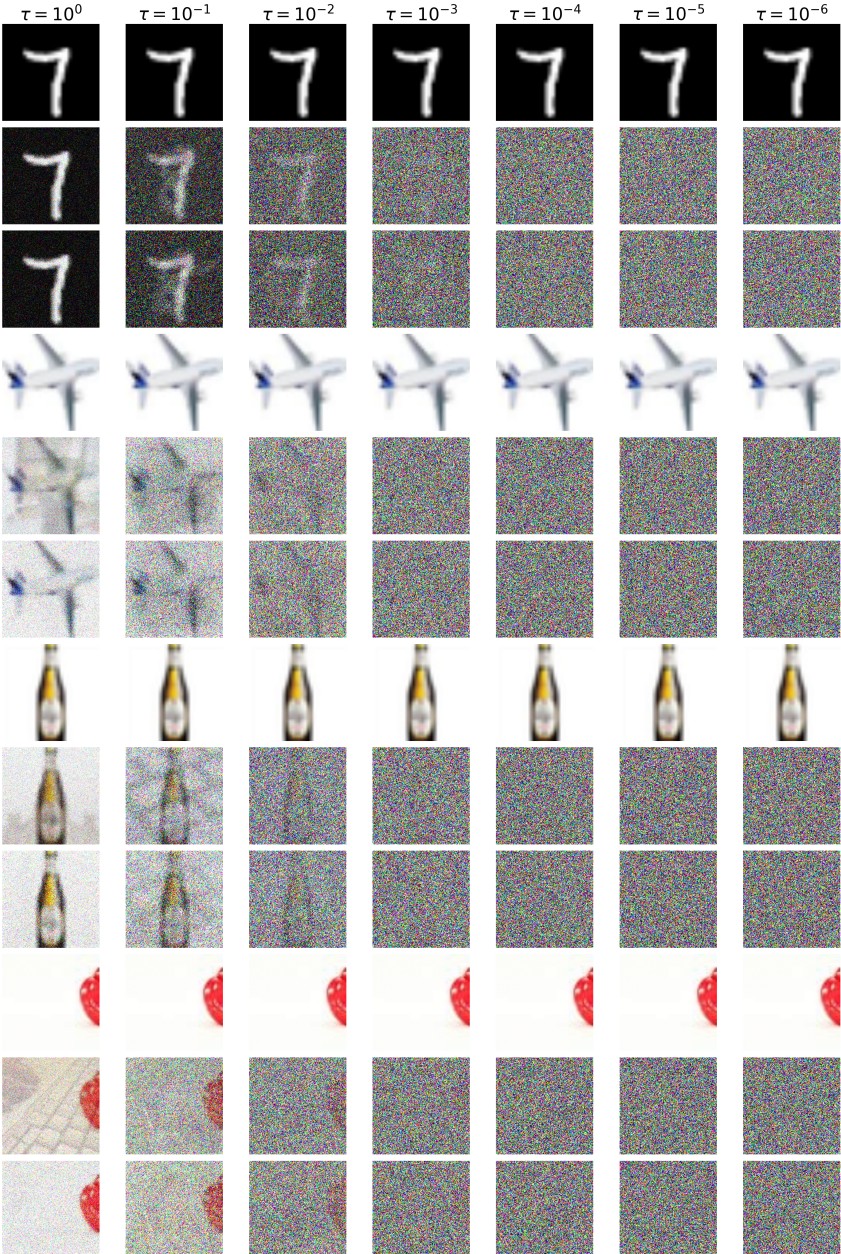

Figure 3: Recovered images under the linear attack for decreasing noise levels $\tau$

## H    THE USE OF LARGE LANGUAGE MODELS

We utilized Large Language Models (LLMs) in three specific ways during this work. First, after conducting a manual review of the state of the art using traditional search engines such as Google Scholar, we used LLMs to assist in identifying additional relevant papers. Second, LLMs were employed to help implement the experiments described in this study. Third, LLMs were used for grammar correction and minor improvements to the flow of the text. Importantly, LLMs were not used to generate or write any paragraphs; their role in writing was limited to minor edits and enhancements.

