# OpenReview forum: "Augmented Mixup Procedure for Privacy-Preserving Collaborative Training"
_ICLR.cc/2026/Conference — Submitted to ICLR 2026_

### Official Review · Reviewer_9hNT · 2025-10-23

**Soundness:** 2
**Presentation:** 2
**Contribution:** 2
**Rating:** 2
**Confidence:** 4

**Summary:**

This paper addresses a security flaw in InstaHide, a privacy-preserving method for collaborative machine learning. The authors demonstrate that InstaHide's process of mixing private and public images can be reversed by an adversary. As a solution, they propose a new method called Singularized Mixup. The core contribution is a procedure where a user's private image is mixed with a weighted average of other *noisy* private images from their own dataset. This deliberate injection of coupled noise before mixing creates a mathematically ill-conditioned inverse problem, making it computationally infeasible for an attacker to reconstruct the original data, thereby securing privacy while largely preserving the model's training accuracy.

**Strengths:**

- Introduces a mathematically simple privacy-preserving mechanism that does not rely on external public datasets.
- The algorithm maintains high model utility, demonstrating only a minor drop in accuracy in exchange for strong privacy guarantees.

**Weaknesses:**

The paper shows empirically that accuracy is preserved, but lacks a theoretical justification for *why*. Could the authors provide a formal proof explaining how gradients can effectively de-mix the feature signals during backpropagation, ensuring the model learns true features rather than spurious correlations from the noise and other mixed images?

The mixup procedure's core assumption of a linear data manifold can fail for complex datasets. Could the authors justify why their method is robust to this? Specifically, how does it prevent the model from learning erroneous features when interpolations between points fall into semantically meaningless regions of the feature space?

The hyperparameters `k` and `r` are currently tuned empirically. Could the authors provide a principled guideline for selecting these parameters for an arbitrary new dataset? A framework that connects a dataset's geometric properties to the choice of `k` and `r` to achieve a predictable privacy-utility trade-off would be a significant contribution.

The paper assumes the adversary is limited to linear attacks. It completely ignores the possibility of more sophisticated, non-linear attacks. For example, a powerful adversary could potentially train a dedicated neural network (like a GAN or a diffusion model) to "de-noise" or "de-mix" the images.

The security analysis doesn't answer a crucial question: For a given image size and a chosen noise radius r, what is the minimum computational cost for an attacker to achieve a certain reconstruction accuracy? Without this, it's hard to anoint the system as "secure" in a practical cryptographic sense; it's more of a demonstration of computational difficulty.

The evaluation on CIFAR and MNIST is promising, but doesn't show scalability. Could the authors evaluate their method on ImageNet to demonstrate if the privacy-utility trade-off holds on a larger, more complex benchmark without a prohibitive drop in accuracy?

The paper contains numerous typos and formatting errors (e.g., `[?]`, "ALgorithm," "miuxp"). A thorough proofread is strongly recommended.

**Questions:**

* Could the authors provide a formal proof for how gradients de-mix feature signals during backpropagation to preserve model utility?
* How does the algorithm prevent learning erroneous features when the linear interpolation between two data points falls into a semantically meaningless region of the feature space?
* Can the authors provide a principled guideline for selecting the hyperparameters `k` and `r` on a new dataset to achieve a predictable privacy-utility trade-off?
* How robust is the algorithm against a non-linear adversary who uses a generative model to learn the de-mixing function?
* What is the minimum computational cost for an attacker to achieve a meaningful reconstruction of a private image?
* Would the stated privacy-utility trade-off hold if the method were evaluated on a large-scale, complex benchmark like ImageNet?

---

> ### Author Response · Authors · 2025-11-26
> **Theoretical clarifications**
>
> **Q1**
> Singularized Mixup preserves the supervised signal even with large perturbations because the gradients remain unbiased and aligned with those of the clean objective. For cross-entropy, linearity in the label yields
> $$\mathcal{L}(\tilde x_i,\tilde y_i)=w_{i1}\mathcal{L}(\tilde x_i,y_i)+w_{i2}\mathcal{L}(\tilde x_i,y_{\pi(i)}),$$
> so
> $$\nabla_\theta\mathcal{L}(\tilde x_i,\tilde y_i)=w_{i1}\nabla_\theta\mathcal{L}(\tilde x_i,y_i)+w_{i2}\nabla_\theta\mathcal{L}(\tilde x_i,y_{\pi(i)}).$$
> Because the noise $e_i$ is independent and mean-zero,
> $$\mathbb{E}\_e[\nabla_\theta \mathcal{L}(x+e,y)] = \nabla_\theta  \mathbb{E}\_e[\mathcal{L}(x+e,y) ],$$
> so SGD estimates unbiased gradients of a smoothed loss whose gradients remain aligned with those of the clean supervised objective in expectation. Noise adds no coherent direction: from $\tilde x_i=w_{i1}x_i+w_{i2}(x_{\pi(i)}+e_i)$ and $e_i\perp\mkern-10mu\perp(x_i,y_i)$,
> $$\mathbb{E}\_e[\nabla_\theta \mathcal{L}(w_{i2}e_i,\tilde y_i)]=0.$$
> Thus only the supervised components accumulate; noise cancels out. Theorem 4.1 shows that the injected noise imposes a linear in $r$ lower bound on any estimator’s reconstruction error, while its label-independence keeps the training gradient unbiased.
>
> **Q2**
> Our method does not assume a linear data manifold. The mixed sample need not be realistic; optimization depends solely on the decomposed gradients, which remain unbiased regardless of whether $\tilde x_i$ falls off-manifold. The SNR bound of Theorem 4.2,
> $$\mathrm{SNR}=\frac{\mathbb{E}\|w_{i1}x_i\|^2}{\mathbb{E}\|(1-w_{i1})(x_{\pi(i)}+e_i)\|^2}\le \tau,$$
> ensures the off-manifold component is dominated by noise and cannot form stable features. Since $w_{i1}>0$, the update always includes
> $$w_{i1}\nabla_\theta \mathcal{L}(\tilde x_i,y_i),$$
> which anchors learning to the true manifold. Thus Singularized Mixup is robust even when linear interpolations leave the data manifold.
>
> **Q3**
> We remove $k$ as a tunable parameter: prior work (Carlini's attack) show $k>2$ weakens security by producing more invertible mixing systems. We fix $k=2$, the hardest-to-invert case. For noise magnitude $r$, we now provide a principled rule based on dataset geometry. Let $D=\mathbb{E}\|X-X'\|$ and $V=\mathbb{E}\|X\|^2$. Setting
> $$r=m_f\cdot D,$$
> Theorem 4.2 shows the privacy target $\mathrm{SNR}\le\tau$ is achieved when
> $$m_f\ge\sqrt{\frac{1}{2c}\left(\frac{\alpha^2}{\tau(1-\alpha)^2}-1\right)},
> \qquad c=\frac{D^2}{2V}.$$
> Thus $(D,V)$ and the desired privacy level $\tau$ uniquely determine $r$, giving a fully principled privacy–utility trade-off.
>
> **Q4**
> We evaluate a strong non-linear adversary: a U-Net trained end-to-end to invert the encoding. This model can approximate highly flexible de-mixing functions. Empirically, once the SNR constraint from Theorem 4.2 is satisfied, the U-Net outputs collapse to blurred/noise-like artifacts, suggesting that the interference term overwhelms the recoverable structure for this class of nonlinear attackers. While not a universal impossibility result, this provides strong evidence that the SNR condition is effective in practice beyond linear attacks.
>
> **Q5**
> We provide an empirical upper bound on attacker capability via the same U-Net inversion attack. This approach is far more computationally expensive than analytic solvers, requiring GPU training on many encoded samples. Even so, no meaningful reconstruction is achieved when $\tau\le 10^{-4}$, showing that an attacker must exceed the recommended privacy level by several orders of magnitude before any signal becomes recoverable. We do not claim cryptographic hardness but show practical infeasibility under standard budgets.
>
> **Q6**
> To study scalability, we evaluate on Tiny-ImageNet, a common proxy for ImageNet due to its diversity and complexity. At a strong privacy setting ($\tau=10^{-6}$), our method reaches 72.50% accuracy while satisfying the SNR bound. This matches or exceeds the strongest InstaHide configurations. Tiny-ImageNet’s greater complexity relative to CIFAR indicates that Singularized Mixup scales favorably, with the main barrier to full ImageNet being compute rather than algorithmic limitations.
>
>
> We thank the reviewers for raising these broader questions. Across these points, we note that extending the theoretical analysis to fully general non-linear adversaries and evaluating on larger-scale datasets such as full ImageNet are natural and important directions. Our results provide strong evidence that the SNR condition governs both reconstruction hardness and utility, and we look forward to exploring these settings in future work.

---

### Official Review · Reviewer_h7YU · 2025-10-26

**Soundness:** 3
**Presentation:** 3
**Contribution:** 2
**Rating:** 4
**Confidence:** 3

**Summary:**

This paper proposes a procedure for training on private data using mixup. Specifically, they identify that prior works that attempt that leverage mixup were broken by follow up works due to the linear system generated by the mixup procedure. To mitigate this vulnerability, the paper proposes a novel procedure called "Singularized Mixup". This method modifies the standard mixup process significantly: each generated sample is a convex combination containing exactly one 'clean' private image. The remaining $k-1$ components are constructed by taking other images from the private dataset and adding substantial, independently sampled noise vectors to them before performing the weighted summation. Crucially, the noise added to a sample is scaled by the same weight used for the sample itself in the mix, coupling the noise structure to the mixing process. Through theoretical and experimental analysis, they show that this change makes it harder for adversaries to recover the private inputs.

**Strengths:**

1. The paper is well-written and easy to follow
2. I like the security analysis that assumes a strong attacker who has access to the mixing matrix.

**Weaknesses:**

1. **Experiments:** Experiments are mostly limited to toy datasets (MNIST, CIFAR-10, CIFAR-100). Utility of the method for more realistic datasets remain unknown.
2. **Potential adaptive attack:** The authors correctly assume a strong adversary model where the mixing matrix $W$ is known (Section 4). This knowledge trivially allows the adversary to perfectly cluster all mixed samples $\tilde{x}$ that contain a specific original image $x_i$, regardless of the added noise $e$. While the paper argues that the "Singularized" nature prevents recovery by averaging (as the average includes other clean images $x_a, x_b, ...$), this overlooks the possibility of a two-stage attack:

    a. Noise Reduction via Averaging: The adversary can average the identified clusters. This averaging process will reduce the variance of the explicit noise components ($e$) due to their independent sampling across different mixups, even if it creates new, complex mixtures of the other clean images.

    b. Blind Source Separation (BSS): The result of Stage 1 is a new set of mixed observations where the explicit noise is significantly attenuated, leaving primarily mixtures of the original image signals. This precisely forms a Blind Source Separation problem. An adversary could potentially apply standard BSS techniques (e.g., Independent Component Analysis - ICA) to this "denoised" system to recover the original images $X$.

    The paper's security analysis focuses on the difficulty of solving $\tilde{X} = WX + E$ directly due to the large, coupled noise $E$. However, it does not appear to address the possibility that an attacker might first effectively remove $E$ through averaging clusters (enabled by knowing $W$) and then solve the remaining BSS problem. This potential attack vector warrants further investigation and discussion by the authors.

**Questions:**

1. Can you address the weakness regarding adaptive attacks?
2. How well does the proposal work for more realistic datasets like imagenet?

---

> ### Author Response · Authors · 2025-11-26
> **Clarifications on Adaptive Attacks**
>
> **Q1.**
> We appreciate the reviewer’s proposed adaptive attack (clustering → averaging → BSS). We first emphasize that our method intentionally uses k=2, which is also InstaHide’s primary setting and, as shown by Carlini et al., the most secure. Increasing k makes inversion *easier*, not harder. Our theoretical analysis already covers this case and formally lower-bounds the adversary’s reconstruction accuracy.
> Importantly, our security model *already* assumes a maximally powerful attacker who can perfectly identify every encoded sample containing a given private image. Thus we directly analyze the attack under this assumption.
>
> 1. **Averaging cannot reduce noise because the noise is tied to the mixing weights.**
> Each encoded sample contains (i) the target image scaled by a random weight, (ii) a *different* interfering image, and (iii) a noise term whose scale is itself multiplied by the same random weight. Within any cluster of samples sharing a private image, both the interfering component and the noise magnitude vary unpredictably. Noise is therefore not independent and not identically distributed; the attacker never receives repeated mixtures of the same sources. Classical variance reduction does not apply, so averaging cannot “wash out’’ noise or isolate the private image. This is precisely reflected in our minimax lower bound: a reconstruction error proportional to the noise radius is information-theoretically unavoidable.
>
> 2. **BSS/ICA assumptions are fundamentally violated.**
> Blind Source Separation techniques require several conditions: (i) multiple mixtures of the *same* set of sources, (ii) noise that is purely additive and independent of mixing, and (iii) consistent sources across observations. None of these hold. Because each pair of images is mixed only once, the attacker never receives repeated mixtures of the same underlying sources. The noise magnitude depends on the random mixing weight, breaking the independent-noise assumption. And even within a perfect cluster, the second mixed image changes from sample to sample, so there is no stable source set to separate. Thus the attacker does *not* obtain a system to which ICA/BSS can be meaningfully applied.
>
> 3. **Nonlinear attacks empirically confirm the same barrier.**
> To validate that these issues are not merely limitations of linear BSS, we evaluated a high-capacity nonlinear U-Net trained end-to-end to invert the encoding. This model is far more expressive than any averaging or BSS-based method, and can in principle learn complex, nonlinear inverse transformations. Despite this, once the injected noise reaches even moderate strength, reconstructions collapse to blurred or noise-like outputs. This matches our theory: the attacker never receives mixtures that satisfy the structural requirements needed to consistently recover the original image. Both linear and nonlinear attacks fail under the same constraints.
>
> **Q2.**
> We expanded our experiments to Tiny-ImageNet, a standard proxy for ImageNet that preserves much of its complexity while being computationally feasible.
> The moderate drop in our original results stemmed mainly from using a weaker classifier rather than from the mixing procedure. In the revised experiments, we align the architecture with that used in InstaHide for a fair comparison. Under this improved setup, our approach experiences only a small reduction in accuracy relative to InstaHide, even at the stricter privacy level used in our updated analysis.
>
> A key addition in the revision is the introduction of a principled privacy parameter τ, which directly determines the required noise magnitude through our theoretical SNR guarantee. Unlike earlier methods that tuned noise empirically, τ provides a formal and transparent privacy–utility trade-off grounded in our theorems.
>
> Updated accuracy results (τ = 10⁻⁶):
>
> - MNIST: ours 99.32% vs. InstaHide 99.66%
> - CIFAR-10: ours 90.51% vs. InstaHide 91.20%
> - CIFAR-100: ours 75.99% vs. InstaHide 74.01%
> - Tiny-ImageNet: ours 72.50% (no InstaHide result available)
>
> These results show that our singularized mixup retains strong utility on substantially more challenging datasets while operating in a formally defined high-privacy regime. They also provide evidence that the method scales effectively to more realistic image distributions, and that the primary computational barrier to full ImageNet evaluation is resource-related rather than methodological.
>
> Thank you for the thoughtful feedback and for helping us improve the clarity of our work.

---

### Official Review · Reviewer_xaSY · 2025-10-31

**Soundness:** 2
**Presentation:** 3
**Contribution:** 3
**Rating:** 6
**Confidence:** 4

**Summary:**

The paper proposes Singularized Mixup (S-Mixup), a modification of Mixup/InstaHide that adds per-term noise before forming convex combinations so that each encoded sample includes exactly one private image and k−1 noisy components, aiming to make the linear inverse problem ill-conditioned for attackers. The authors analyze security under an isotropic Gaussian assumption and evaluate on MNIST/CIFAR-10/100.

**Strengths:**

- Clear motivation: repeated private-sample reuse in InstaHide is a key attack vector, and S-Mixup avoids that by design.
- The paper is well structured in terms of explaining the motivation and the vulnerabilities of the prior methods.
- There are sufficient experiments to demonstrate the properties of the proposed method.

**Weaknesses:**

## Weaknesses
- The major issue with this method would be the significant classification accuracy dropping when reasonable noise levels are used. For mf=1,2,4 the recovered images could be simply denoised by a learned image restoration network to obtain good results. For mf=8,16,32 the performance drop is significant. This severely limits cases in which the proposed method can be used.
- The theorem has too strict assumptions (gaussian for data etc.) and the connection to the experimental work is left a bit tenuous. I am not sure it provides meaningful insight in understanding or evaluating the proposed method. Given the assumptions on the data don't hold for the experimental part, the only value could be in understanding the characteristics of the method which is also not very well explained. The connection between the theory, proposed method, and empirical results should be strengthened and explained more clearly.

## Minor Issues
There are a couple of grammatical issues throughout the paper.
- Typo in line 330 "ALgorithm"
- Typo in line 330 "miuxp"
- Line 580 "coorsinates"
    etc.

Line 294 the reference is not rendered.

**Questions:**

- In a multi-party scenario where each party has k (for example 1000) private labeled images, when would using S-Mixup across n parties improve performance over training with standard Mixup on private data? This seems like an interesting case to analyze. This would demonstrate an interesting tradeoff between secure training on more data versus using minimal clean data.
- Can the matrix W be constructed to make the system more ill-conditioned, rather than sampled at random?
- The paper states that S-Mixup makes the inverse problem ill-conditioned. Can one estimate or bound the condition number as a function of the noise level?

---

> ### Author Response · Authors · 2025-11-26
> **S-Mixup in Federated Settings**
>
> **Q1.**
> The multi-party setting indeed highlights the tradeoff between (i) preserving each party’s privacy and (ii) improving accuracy by effectively training on a larger combined dataset.
>
> In our federated CIFAR-10 experiment, each party holds a disjoint local subset of the data and independently applies S-Mixup before sending only the mixed samples to the central server. Table 4 reports the following results:
>
> | Number of Parties | Baseline (Local Only) | S-Mixup Union (All Parties) |
> |-------------------|------------------------|------------------------------|
> | 10                | 87.25%                 | 90.63%                       |
> | 20                | 85.56%                 | 90.32%                       |
> | 30                | 84.08%                 | 88.23%                       |
>
> (see Table 4 in the paper) :contentReference[oaicite:0]{index=0}
>
> These results directly illustrate when S-Mixup across \(n\) parties improves performance over training with standard Mixup locally.
>
> ---
>
> **Q2.**
> It is possible to construct
> $$
> W = D_{1} + D_{2}P_{\pi}
> $$
> to be more ill-conditioned than when sampling weights at random. For example, selecting many rows with $w_{i1} \approx w_{i2}$ introduces near-linear dependencies and reduces $\sigma_{\min}(W)$, while choosing permutations with long cycles can further cluster singular values. Both choices increase $\kappa(W)$.
>
> We do not follow such deterministic constructions because excessive ill-conditioning harms utility: when $w_{i1}$ and $w_{i2}$ become nearly equal, the encoded sample carries weak signal, making training unstable. Our design therefore constrains $\|w_i\|_\infty \le \alpha$, ensuring that mixtures remain informative while still creating enough randomness to destabilize inversion. Random bounded weights achieve a good balance between privacy and utility without requiring adversarial tuning of $W$.
>
> ---
>
> **Q3.**
> Because $ W = D_{1} + D_{2}P_{\pi} $ differs from the permutation matrix $P_{\pi}$ by
> $$
> \Delta = D_1 + (D_2 - I)P_\pi ,
> $$
> and because $\|D_1\|\_2 \le \alpha$ and $\|D_2 - I\|\_2 \le \alpha$, we have
> $$
> \|\Delta\|\_2 \le 2\alpha.
> $$
> By standard singular–value perturbation bounds (Weyl/Mirsky),
> $$
> |\sigma_j(W) - \sigma_j(P_\pi)| \le \|\Delta\|\_2 \le 2\alpha,
> $$
> and since $P_\pi$ has singular values all equal to $1$,
> $$
> 1 - 2\alpha \\le \sigma_{\min}(W) \le \sigma_{\max}(W) \\le 1 + 2\alpha,
> $$
> Therefore,
> $$
> \kappa(W) = \frac{\sigma_{\max}(W)}{\sigma_{\min}(W)}
> \le \frac{1 + 2\alpha}{1 - 2\alpha}.
> $$
>
> Noise enters through $E = D_{2}E_{\text{noise}}$. If $\|E_{\text{noise}}\|\le r$, then $\|E\|\le r$ because $\|D_2\|\le 1$. The adversary’s reconstruction error satisfies
> $$
> \|W^{-1}E\| \le \|W^{-1}\|\,\|E\|
> \le \frac{r}{1 - 2\alpha}.
> $$
>
> This shows explicitly how conditioning interacts with noise: smaller $\alpha$ increases $\sigma_{\min}(W)$ and decreases noise amplification, while larger $\alpha$ pushes $\sigma_{\min}(W)$ toward zero and makes the inverse problem increasingly ill-conditioned.
>
> ---
>
> Thank you for the constructive questions.

---

### Official Review · Reviewer_L4sq · 2025-11-12

**Soundness:** 2
**Presentation:** 2
**Contribution:** 2
**Rating:** 2
**Confidence:** 4

**Summary:**

The paper addresses the vulnerabilities of image encryption methods, such as InstaHide, to adversarial attacks through a simple but effective modification to the mixup procedure. The contribution is based on the insight that introducing noise before the mixing operation makes private data reconstruction significantly less effective compared to adding it after the mixup operation. Through a thorough review of systematic attacks to which InstaHide is susceptible, the authors demonstrate that their method is resilient and provide a formal proof of its security.

**Strengths:**

The paper addresses the vulnerabilities of image encryption methods, such as InstaHide, to adversarial attacks through a simple but effective modification to the mixup procedure. The contribution is based on the insight that introducing noise before the mixing operation makes private data reconstruction significantly less effective compared to adding it after the mixup operation. Through a thorough review of systematic attacks to which InstaHide is susceptible, the authors demonstrate that their method is resilient and provide a formal proof of its security.

**Weaknesses:**

1. In page 8, the paper claims ”for more complex data, such as CIFAR-10
and CIFAR-100, a noise factor of mf = 8 is sufficient to prevent the recovery
of the original image.” This seems to be based on the results of Figure 2
rather than theory. This makes it difficult for users to apply their method
to new datasets, as it’s hard to determine what the appropriate mf may
be sufficient to prevent reconstruction of the original data. Can the authors please clarify?

2. The experiments only explored the image domain.

3. The experiments covered MNIST, CIFAR-10, and CIFAR-100, but do not
explore dealing with medium to large datasets (IMAGENET), so it is empirically unclear if the method works for these larger systems. CIFAR-100, the hardest of the three datasets, has the largest drop in accuracy with increasing scaling factor mf. Would this be more pronounced with even more difficult datasets to the extent that their algorithm becomes impractical?

4. Figure 2, page 9: Wouldn’t the effect of mf be more apparent if every
row contained the same image? Is it possible that the effect of mf varies
depending on the individual characteristics of each image?

5. In page 7, line 330, capitalization of L in ”ALgorithm. ” Missing citation in page 6, eq. 9.

**Questions:**

See Weakness.

---

> ### Author Response · Authors · 2025-11-26
> **Noise selection theory**
>
> **Weakness 1.**
> In the original submission, guidance for choosing the noise multiplier was mostly empirical. This made it unclear how practitioners should tune the noise level on new datasets. In the revision, we address this by introducing two theorems that together provide a principled, dataset-dependent procedure.
>
> Theorem 4.1 explains *why* increasing the noise multiplier improves security: any reconstruction attempt must incur an error that scales linearly with the noise norm. This applies to all estimators—linear or nonlinear—and therefore captures a fundamental barrier to inversion.
>
> Theorem 4.2 gives the *actionable recipe*. It connects the required noise magnitude to the expected signal-to-noise ratio (SNR) of each encoded sample. SNR measures how much information about the private image survives mixing; ensuring $\text{SNR} \le \tau$ guarantees that no meaningful visual structure is leaked.
>
> Most importantly, Theorem 4.2 yields an explicit formula for the minimum noise multiplier needed to satisfy a chosen privacy level $\tau$:
>
> $$
> m_f \\ge\\
> \sqrt{
> \frac{1}{2c}
> \left(
> \frac{\alpha^2}{\tau (1-\alpha)^2}
> \-\
> 1
> \right)
> }
> $$
>
> where
> • $\alpha$ is the upper bound on the mixing weights,
> • $\tau$ is the privacy target (smaller means stronger privacy),
> • $c = D^2/(2V)$, with
>  – $V = \mathbb{E}\|X\|_2^2$ (average squared norm of samples),
>  – $D = \mathbb{E}\|X - X'\|_2$ (average distance between two samples).
>
> Both $D$ and $V$ can be estimated efficiently from a small data subset, meaning the practitioner can compute $m_f$ *directly* without visual inspection or hyperparameter guessing. As a result, the revised paper replaces heuristic dataset-specific claims such as “$m_f = 8$ works for CIFAR-10’’ with a general rule grounded in theory.
>
> **Weakness 2.**
> It is correct that our experiments focus on images. This matches InstaHide, our main point of comparison, which is itself an image-only obfuscation method. Nevertheless, we agree that applying Singularized Mixup to other modalities—such as text, audio, or tabular data—is an important direction. We added an explicit remark in the conclusion noting this as future research.
>
> **Weakness 3.**
> To address concerns about performance on more realistic datasets, we expanded our evaluation to include Tiny-ImageNet, which is considerably more complex than CIFAR-100 and commonly used as a practical proxy for ImageNet.
> We also determined that the accuracy drops in the original submission were mainly due to underpowered classifier architectures. In the revised version, we employ models with capacity comparable to those used in InstaHide, leading to substantially improved results.
> At the strict privacy level $\tau = 10^{-6}$, our method matches or exceeds InstaHide on CIFAR-100, and achieves 72.50% accuracy on Tiny-ImageNet. This demonstrates that the method remains highly practical even under conservative noise levels derived from our SNR theory.
>
> **Weakness 4.**
> We fully agree that reconstructions across different noise magnitudes should use the same underlying image. In the revision, Figures 1 (linear attack) and 2 (nonlinear U-Net attack) now use the *same* CIFAR-10 sample for all noise levels $\tau$. For each $\tau$, we show the *best recovered image* to represent the strongest possible attacker.
> Appendix G further includes full reconstruction grids for multiple images across all $\tau$ values. This makes the influence of $m_f$ and the SNR threshold clearly interpretable.
>
> Thank you for the thoughtful and constructive feedback, which helped us significantly strengthen both the theoretical and empirical components of the paper.

---

### Author Response · Authors · 2025-11-27
**Summary**

We thank the reviewers for their careful reading, constructive feedback, and insightful suggestions, which have greatly improved our paper. We revised the manuscript to address all conceptual and experimental concerns. Below, we summarize the major changes and explain how they respond to the reviewers’ main comments.

---

# 1. Removal of the parameter $k$
A central change in the revised paper is the complete removal of the mixup-size parameter $k$.
Initially, the algorithm mixed $k$ private samples plus noise. We noted that:
- Increasing $k$ reduces security (as observed in Carlini et al.), because larger $k$ produces a
more structured linear system that becomes easier to invert.
- InstaHide’s own strongest experimental setting mixes 2 private and 2 public images, not large
$k$.

In the new algorithm, each encoded sample mixes **exactly two private images**:
$$
\tilde{x}\_i = w_{i1} x_i + w_{i2}(x_{\pi(i)} + e_i),
$$
eliminating the vulnerabilities induced by repeated reuse of many private samples.
This simplification both strengthens security and makes the mechanism easier to apply in practice.

---

# 2. Two new theorems that formally characterize privacy and guide the choice of noise

A major limitation noted by reviewers was that the noise multiplier $m_f$ was previously chosen
**empirically**, making it difficult for practitioners to tune privacy–utility trade-offs or transfer the
method to new datasets.
In the revised version, Section 4 introduces **two new theorems (Theorem 4.1 and Theorem 4.2)** that
provide a rigorous framework for setting the noise level.

**Theorem 4.1 — Lower bound on reconstruction error**
This theorem proves formally that any adversary attempting to invert the system
$$
\tilde{X} = W X + E
$$
faces a minimum expected reconstruction error that scales linearly in the noise norm $r$:
$$
\mathbb{E}\|x_i - \hat{x}_i\|_2^2 \\ge\\ r^2\, T_i.
$$

**Theorem 4.2 — A principled, theoretically grounded method for selecting $m_f$**
We now set
$$
r = m_f \cdot D,\qquad D=\mathbb{E}\|X - X'\|
$$
and provide a theorem guaranteeing that the
$\mathrm{SNR}$ remains below a prescribed privacy parameter $\tau$ if and only if
$$
m_f \\ge
\sqrt{\frac{1}{2c}\left(\frac{\alpha^2}{\tau(1-\alpha)^2}-1\right)} .
$$

This addresses the reviewers’ concerns directly:
- $m_f$ is no longer heuristic,
- its role is theoretically justified,
- the privacy–utility trade-off is explicit and user-controllable through $\tau$.

---

# 3. New experiments on Tiny-ImageNet and higher-capacity models

We added Tiny-ImageNet experiments and a higher-capacity ResNet-50 based classifier, matching those used in the InstaHide paper to ensure a fair comparison.

**Results**
Accuracy at strong privacy level (\(\tau=10^{-6}\)):
| Method | MNIST | CIFAR-10 | CIFAR-100 | Tiny-ImageNet |
|--------|--------|-----------|------------|----------------|
| InstaHide (k=4) | 99.66 | 91.20 | 74.01 | – |
| **Ours** | 99.32 | 90.51 | **75.99** | **72.50** |

Our revised method now matches or exceeds InstaHide’s accuracy on all datasets while using
stricter noise.

This directly addresses reviewer concerns on scalability and practical utility.

---

# 4. New federated-learning experiments

Reviewers explicitly asked about the multi-party scenario where each party has limited private
data.  The new version includes experiments showing that training on the union of singularized mixup datasets consistently outperforms training on small clean subsets, confirming that S-Mixup enables effective and secure collaborative training.

---

# **Overall Impact of the Revisions**
The new submission introduces several conceptual, theoretical, and experimental improvements:

1. **Theoretical foundations are now rigorous**
   - Formal lower bounds on reconstruction error
   - A principled rule for selecting noise magnitude
   - A transparent and tunable privacy parameter $\tau$

2. **Experiments are more comprehensive, realistic, and fair**
   - Added Tiny-ImageNet
   - Upgraded models to match InstaHide’s experimental setup
   - Added federated learning experiments
   - Linear and nonlinear (U-Net) attackers evaluated at multiple $\tau$ levels
   - Security holds even at $\tau=10^{-4}$; accuracy evaluated at the stricter $\tau=10^{-6}$

3. **Empirical findings now align with theoretical expectations**
   - Reconstruction SNR decreases as predicted by Theorem 4.2. Importantly, this theorem concerns the SNR of the *encoded* sample, not of the adversary’s reconstruction. Our experiments show that whenever the encoded-sample SNR falls below the theoretical threshold $\tau$, all reconstruction attempts—linear and nonlinear—fail, empirically confirming the predictive power of the theoretical bound.

   - Utility decreases very slowly even at security-level noise
   - Visual recovery is impossible even with powerful nonlinear attacks

---

### Comment · Area_Chair_xwF4 · 2025-11-28
**A gentle reminder to participate in the author–reviewer discussion.**

Dear Reviewers,

Thank you once again for your service to ICLR 2026. Now that the authors have submitted their rebuttal, could you please engage in the interactive discussion with them? Your participation would be very helpful to the authors, and they would greatly appreciate it. Please also read the authors’ response together with the other reviews and consider whether the rebuttal or any additional comments influence your assessment of the paper.

Thank you again for your efforts.

Best wishes,

Your AC

---

> ### Author Response · Authors · 2025-11-28
> **Clarification**
>
> Dear AC,
>
> Given the recent OpenReview bug, could you please clarify whether we should expect reviewers to respond anymore?
>
> Thank you,
> Authors

---

### Meta-Review · Area_Chair_jfGi · 2026-01-04

**Summary:**

My suggested decision to Reject is based on the synthesis of four reviews. While the authors provided a substantial rebuttal that effectively addressed concerns regarding heuristic parameter selection and experimental scope (adding Tiny-ImageNet and Federated Learning), fundamental concerns regarding the theoretical foundation remain. Specifically, reviewers (xaSY, 9hNT) noted that the new theoretical analysis relies on idealized assumptions (e.g., isotropic Gaussian data) that do not translate rigorously to real-world image distributions. Furthermore, the paper lacks a formal information-theoretic security proof against general nonlinear adversaries, relying instead on empirical demonstrations (U-Net failure). Consequently, while the work has significantly improved, the gap between theory and practice prevents a clear acceptance at this stage.

**Reviewer Concerns:**

Effectively Addressed Concerns:
1. Heuristic Parameters (L4sq, 9hNT): The introduction of Theorems 4.1 and 4.2 successfully shifted the noise selection from an empirical guess to a principled framework based on a privacy parameter (τ).
2. Limited Experimental Scope (h7YU, 9hNT, L4sq): The addition of Tiny-ImageNet experiments, higher-capacity models (ResNet-50), and Federated Learning scenarios directly addressed concerns about scalability and practical utility.
3. Specific Attack Vectors (h7YU, 9hNT): The authors convincingly rebutted the feasibility of the proposed clustering+BSS attack and empirically demonstrated robustness against powerful nonlinear adversaries (U-Net).

Outstanding / Partially Addressed Concerns:
1. Idealized Theoretical Assumptions (xaSY, 9hNT): The theoretical guarantees heavily rely on simplified assumptions (e.g., isotropic Gaussian data). The disconnect between this idealized model and the complex manifold of real image data remains a significant theoretical gap.
2. Lack of Formal Security Proof (9hNT): While the U-Net evaluation is promising, it constitutes an empirical validation rather than a formal, cryptographic, or information-theoretic guarantee against all possible nonlinear inversion strategies.
3. Scalability Limits (9hNT): While Tiny-ImageNet is an improvement, the performance on full-scale ImageNet remains unverified.

**Reviewer Scores:**

Reviewer L4sq (Initial: 2 → Estimated: 4): Would likely improve due to the resolution of the heuristic parameter issue, but may remain borderline due to the remaining theoretical gap.

Reviewer xaSY (Initial: 6 → Estimated: 6): While utility concerns were resolved, the "tenuous connection" between the Gaussian theory and experiments prevents a higher score.

Reviewer h7YU (Initial: 4 → Estimated: 6): Would likely lean towards acceptance as their specific concern regarding adaptive attacks was effectively refuted.

Reviewer 9hNT (Initial: 2 → Estimated: 4): Significant improvements were made, but the lack of a formal proof against nonlinear attacks limits the score increase.

---

### Decision · Program_Chairs · 2026-01-26

Reject